# Application of Multimodal MRI in the Early Diagnosis of Autism Spectrum Disorders: A Review

**DOI:** 10.3390/diagnostics13193027

**Published:** 2023-09-22

**Authors:** Miaoyan Wang, Dandan Xu, Lili Zhang, Haoxiang Jiang

**Affiliations:** 1Department of Radiology, Affiliated Children’s Hospital of Jiangnan University, Wuxi 214000, China; wyflipped@gmail.com (M.W.); xudandan666888@163.com (D.X.); 2Department of Child Health Care, Affiliated Children’s Hospital of Jiangnan University, Wuxi 214000, China

**Keywords:** autism spectrum disorder, early diagnosis, magnetic resonance imaging, glymphatic system

## Abstract

Autism spectrum disorder (ASD) is a neurodevelopmental disorder in children. Early diagnosis and intervention can remodel the neural structure of the brain and improve quality of life but may be inaccurate if based solely on clinical symptoms and assessment scales. Therefore, we aimed to analyze multimodal magnetic resonance imaging (MRI) data from the existing literature and review the abnormal changes in brain structural–functional networks, perfusion, neuronal metabolism, and the glymphatic system in children with ASD, which could help in early diagnosis and precise intervention. Structural MRI revealed morphological differences, abnormal developmental trajectories, and network connectivity changes in the brain at different ages. Functional MRI revealed disruption of functional networks, abnormal perfusion, and neurovascular decoupling associated with core ASD symptoms. Proton magnetic resonance spectroscopy revealed abnormal changes in the neuronal metabolites during different periods. Decreased diffusion tensor imaging signals along the perivascular space index reflected impaired glymphatic system function in children with ASD. Differences in age, subtype, degree of brain damage, and remodeling in children with ASD led to heterogeneity in research results. Multimodal MRI is expected to further assist in early and accurate clinical diagnosis of ASD through deep learning combined with genomics and artificial intelligence.

## 1. Introduction

Autism spectrum disorder (ASD) is a neurodevelopmental disorder characterized by impairments in social communication and interaction, with restricted repetitive stereotyped behaviors [1]. According to data from the United States Department of Education, the risk of developing ASD has increased by 10% to 17% annually [2]. In the United States, 1 in 36 children are diagnosed with ASD, and the prevalence is 3.8 times higher in boys than in girls [3]; the disorder places a heavy financial and emotional burden on families and society [4,5]. ASD is usually diagnosed at the age of 4 years, and magnetic resonance imaging (MRI) can aid early detection of abnormal changes in the brain [6], including increased brain volume, impaired integrity of white-matter fiber tracts, and abnormalities in the connectivity of the brain’s structural and functional networks, tissue perfusion, and neuronal metabolism [7,8,9]. With early diagnosis of ASD, scientific and effective interventions may help remodel the neural connectivity of the brain and improve the quality of life in children with ASD [10]. Currently, widely used MRI techniques include structural MRI (sMRI), diffusion tensor imaging (DTI), functional MRI (fMRI), three-dimensional arterial spin labeling (3D-ASL), and proton magnetic resonance spectroscopy (^1^H-MRS). This review aimed to explore potential imaging biomarkers to assist in early clinical diagnoses and precise interventions by analyzing the role of multimodal MRI in ASD from the existing literature (Figure 1).

## 2. Materials and Methods

A literature search of relevant databases (PubMed and Web of Science) was conducted to identify articles published between January 2010 and May 2023, using the following keyword strategy: “autism spectrum disorder” AND (“structural MRI” OR “diffusion tensor imaging” OR “functional MRI” OR “arterial spin labeling” OR “^1^H-MRS”) AND “diagnosis” AND “children”. Data were included from children (1) who had been diagnosed with ASD and (2) whose brain MRI had been performed. The exclusion criteria were as follows: (1) review articles, letters, comments, and case reports; (2) age > 6 years; (3) no relevant data; (4) studies using animal models; (5) genetic research (Figure 2).

## 3. Research Progress of Structural MRI in ASD

In sMRI, changes in brain volume in children with ASD are shown using voxel-based morphometry. Changes in the surface area and thickness of the cerebral cortex can be studied using surface-based morphometry. The brain volume of children with ASD begins to increase at the age of 12–24 months, rapidly increases at 2.5 years, and increases by approximately 10% compared with the brain of typically developing (TD) children at the age of 2–4 years; the brain volume then increases slowly during late childhood and adolescence [6,13,14,15]. Excessive early brain growth may be associated with an increased number of neurons, which in turn results in an excess of axons, dendrites, synapses, and myelin, leading to increase in both gray- and white-matter volumes in the brain [16]. In addition, the symmetric amplification of germinal cells around the ventricles in individuals with ASD may lead to an increase in minicolumns, which could contribute to the expansion of the cortical surface area. This increased surface area is primarily located in the middle occipital gyrus, cuneus, and lingual gyrus areas, further promoting early brain overgrowth [15,17,18]. Compared with TD children, the volumes of the bilateral superior frontal gyrus, left precuneus, left inferior occipital gyrus, right angular gyrus, bilateral superior temporal gyrus, and left inferior parietal lobule of the brain are increased in children with ASD [6,19,20,21], whereas the right inferior temporal gyrus decreases in volume, thereby reflecting the atypical nature of the brain structure in children with ASD [22]. As the angular and superior frontal gyri are located in the cognitively relevant default mode network (DMN), their volume changes are closely associated with social and cognitive deficits [15]. Children with ASD have increased hippocampal volume compared with that of TD children [23,24,25]. Specifically, the left hippocampal white-matter volume increases [26] while bilateral gray-matter volume decreases [27] or increases compared with TD children [28]. The increase in hippocampal volume may be associated with an increase in pyramidal neurons during the birth process [29]. Due to the involvement of the hippocampus in the core functions of the “social brain,” changes in its volume can lead to language and cognitive impairments [26,27]. Previous studies have shown that histopathological changes in the cerebellum, such as the reduction of granule cells, hypertrophy, and atrophy of cerebellar nuclei can be observed in the postmortem brains of individuals with ASD [30]. Increased cerebellar volume at 4–6 months of age in children with ASD can predict the emergence of restricted and repetitive behaviors in early childhood [31]. Increased cerebellar volume during infancy and childhood in individuals with ASD may be related to early brain overgrowth [30], while the decreased cerebellar volume in adolescents and adults with ASD is positively correlated with the severity of motor restrictions [32,33]. Additionally, asymmetrical changes in brain region volumes exist in individuals with ASD, with a decrease in volume in the left motor system and an increase in volume in the right motor system, leading to motor abnormalities [34]. Rightward asymmetry is observed in the planum temporale and posterior superior temporal gyrus, affecting the language function area known as Wernicke’s area located between them, which is associated with language impairments in individuals with ASD [35].

In addition to changes in volume, children with ASD also exhibit abnormalities in cortical thickness. Children with ASD have significant cortical thickening in early childhood, accelerated thinning in late childhood and adolescence, and gradual cortical thinning with age in adulthood. These cortical changes in the inferior frontal, inferior temporal, and posterior cingulate gyri are the most pronounced. These changes are associated with social cognitive deficits, verbal communication deficits, and stereotypical movements [36,37,38,39,40,41,42,43]. Normally, brain regions maintain structural and functional laterality. Cortical thickness asymmetry of the medial frontal, orbitofrontal, inferior temporal, and cingulate gyri is reduced in children with ASD, reflecting disruption of lateralized neurodevelopment [44,45]. Orbitofrontal cortical abnormalities are strongly associated with self-regulation and social–emotional–behavioral deficits in children with ASD [46]. As the cerebral cortex expands within the limited space in the skull, it gradually increases the number of cortical folds [47]. In children with ASD, the gyrification index, which reflects the degree of cortical folding, increases atypically in childhood and then decreases in adolescence and adulthood, primarily in the frontal and parietal regions [48,49,50]. In children with ASD, the cortical gyrification index is significantly higher in the bilateral temporal lobes, left isthmus cingulate, and left frontal lobe and lower in the right precuneus compared with that in TD children, thereby reflecting the presence of atypical rotational patterns in children with ASD [49,51,52,53]. Gao et al. [54] used sMRI combined with convolutional neural networks and individual structural covariance networks for the early assessment of ASD with an accuracy of 71.8%, sensitivity of 81.25%, and specificity of 68.75%. They also suggested that abnormalities in the prefrontal cortex and cerebellum may be early biomarkers for the diagnosis of ASD. In addition, sMRI not only reveals the morphological differences in brain structures at different ages, but also further clarifies the pathophysiological alterations of the disease by correlating them with the core symptoms of ASD. Thus, sMRI contributes to the early diagnosis of ASD. A summary of the literature on structural MRI-based morphological changes in the brains of individuals with ASD is provided in Table 1.

## 4. Studying Brain Network Changes in Children with ASD Using DTI

Diffusion tensor imaging is a non-invasive technique for assessing the orientation and connectivity of cerebral white-matter fiber bundles, thereby allowing for qualitative and quantitative analyses of water-molecule diffusion characteristics within three-dimensional spaces [51]. Commonly used parameters include fractional anisotropy (FA) and mean diffusion (MD), which reflect microstructural changes in the white matter [55]. FA increases in children with ASD before the age of 4 years, which may be associated with excess prenatal neurons, leading to frontal axonal overconnectivity [56]. This excessive axonal growth leads to signal delays and metabolic inefficiencies in connecting different regions of the brain, thus affecting myelin development [56]. After the age of 4 years, the rate of myelination development slows down, resulting in a decrease in the integrity of white-matter fiber bundles throughout the brain and a gradual decline in FA [57,58]. Compared with TD children, children with ASD show increased FA during early childhood, primarily observed in the corpus callosum, inferior longitudinal fasciculus, inferior fronto-occipital fasciculus, posterior cingulate cortex, and limbic lobe. A higher FA in the corpus callosum is associated with impaired social and communicative functions, and that in the inferior longitudinal fasciculus and inferior fronto-occipital fasciculus is associated with difficulty in recognizing emotions and facial expressions [11,59]. During the childhood phase in ASD individuals, there is a decline in FA, primarily observed in the sagittal stratum, corpus callosum, superior cerebellar peduncle, superior longitudinal fasciculus, cingulum, and uncinate fasciculus, which are associated with motor functions, language, and social impairments [57,58,60,61,62,63,64,65]. However, Weinstein et al. [66] found increased FA values in the corpus callosum and superior longitudinal fasciculus in children with ASD aged 1.5–6 years old. This inconsistency in research findings may be attributable to variations in the age range of the study samples [67]. In comparison with TD children, children with ASD show decreased MD values in the left corpus callosum, posterior cingulate cortex, limbic lobe, and insular cortex during early childhood, which are associated with cognitive impairments [11]. In ASD individuals, during late childhood, adolescence, and adulthood, the increased MD is mainly observed in the left parahippocampal gyrus, left sagittal gyrus, left superior temporal gyrus, and left arcuate fasciculus. The atypical lateralization reflects abnormal connectivity in the left hemisphere white matter and is associated with language and praxis impairments [8,68,69,70,71,72].

Furthermore, graph theory analysis based on DTI can reflect the whole-brain connectivity characteristics of individuals with ASD. Network properties include the clustering coefficient, local efficiency, shortest path length, global efficiency, and small-worldness coefficient. Based on DTI brain network connectivity, the node efficiency of the left pallidum, right caudate nucleus, left precuneus, thalamus, and bilateral superior parietal cortex increased in children with ASD aged 2–6 years compared with TD children [73,74]. Increased node efficiency reflects the presence of hyperconnectivity in the brain structures of preschool-going children with ASD, which may be related to early brain overgrowth. The nodal efficiency of the precuneus is correlated with the severity of ASD [73]. The increased nodal efficiency primarily occurs in the left hemisphere of the brain, specifically in regions associated with language and social communication functions. Therefore, the enhanced network efficiency in the left hemisphere may contribute to language and social impairments in children with ASD [73]. However, reduced nodal efficiency in adolescents and adults with ASD compared with TD individuals suggests that impaired integrity of white-matter fiber tracts may disrupt the topological properties of nodal connectivity fibers. The regions with reduced nodal efficiency are mainly located in the left inferior frontal gyrus, left precentral gyrus, right cingulate gyrus, right precuneus, and right amygdala, which are associated with impaired language and social communication [75,76]. Qian et al. [76] conducted a longitudinal study on ASD children aged 2–5 years and found that there was an increase in four additional hubs at the age of 4–5 years compared with 2–3 years. These additional hubs include the left anterior cingulate and paracingulate, right dorsolateral superior frontal, right middle frontal, and angular gyri. This suggests that the brain in ASD has a remarkable flexibility to rewrite itself [77]. Additionally, children with ASD show a decrease in small-world attributes, indicating a disruption in the balance of information transmission within the white-matter structural network. Shortened path lengths, increased global efficiency, and clustering coefficients of the basal ganglia, limbic, and paralimbic systems in children with ASD indicate brain hyperconnectivity, which in turn is associated with repetitive stereotyped behaviors and learning and memory disorders [74]. However, Li et al. [78] found that monozygotic twins with ASD had reduced global efficiency and increased characteristic path lengths in brain networks, which were associated with core symptoms, such as repetitive behaviors. The differences in findings reflect atypical brain development in children with ASD, which is associated with age, different subtypes, brain damage, and excessive remodeling and requires further research.

In addition, edge density (ED) can further elucidate the brain connectome by examining the potential fiber bundles between cortical nodes. Changes in connectivity around the ventricles are associated with ASD, where ED increases during early childhood. However, during adolescence, there is a widespread reduction in ED within white-matter fiber bundles except for the internal capsule. In adults with ASD, the regions exhibiting decreased ED are mainly located in the posterior commissural and paraventricular white-matter tracts [8]. Furthermore, a machine-learning method that combines the DTI brain network and T1-weighted imaging to assess children with ASD aged 2–6 years has an accuracy, sensitivity, and specificity of 88.8%, 93%, and 83.8%, respectively [79]. Therefore, studies using DTI in children with ASD can not only reveal abnormal developmental trajectories and network connectivity changes in the brain at an early stage, but, upon combination with machine learning, are more helpful for the early diagnosis of children with ASD [80]. Table 2 summarizes the literature on brain network alterations using DTI in patients with ASD.

## 5. Functional MRI–Based Functional Brain Network Alterations in ASD

Resting-state fMRI (rs-fMRI) reflects neural activity and functional changes in the functional areas of the brain by studying the temporal correlation of blood oxygen-level-dependent signals in different brain regions [81]. Based on rs-fMRI, abnormal functional connectivity (FC) was found in the brains of children with ASD, most notably located in the DMN, salience network (SN), central executive control network (ECN), ventral attention networks (VAN), and dorsal attention networks (DAN) [82,83]. The DMN, with the rostral anterior cingulate cortex and medial prefrontal cortex as key brain regions, is primarily involved in intrinsic mental activities, such as recalling the past, envisioning the future, and simulating non-occurring social interactions [84]. The SN, with the anterior cingulate and ventrolateral prefrontal cortices as key brain regions, primarily differentiates between internal and external stimuli to guide behavior [85], and damage to the SN may lead to impairments in social–emotional functioning [86]. The ECN, with the dorsolateral prefrontal and parietal cortices as core regions, participates in processes of attention, decision making, working memory, and response selection [87]. The DAN, composed of the dorsolateral prefrontal cortex, superior temporal gyrus complex, and other areas, is primarily involved in controlling attention processes [88]. In contrast to TD, functional network hyperconnectivity in children with ASD, owing to early brain overgrowth and increased neural density, is located mainly in the medial prefrontal, cingulate, and temporal poles of the DMN, and is associated with socio-emotional disorders [89,90]. Recent studies suggest that early and persistent abnormal connections between the temporal lobe and the cuneus and precuneus lobes during early childhood in patients with ASD may be biomarkers of early language and social dysfunction in children with ASD [91]. There is also hyperconnectivity between brain networks in children with ASD, mainly between the DMN and DAN, DMN and control networks, and visual and sensorimotor networks [92,93,94]. With age, adolescents and adults with ASD mainly exhibit the coexistence of hypo- and hyperconnectivity networks, reflecting uneven brain development. Hypoconnectivity is mainly existent in the sensorimotor brain regions, and hyperconnectivity in the prefrontal and parietal cortices [83,89,90,95,96,97,98,99,100,101,102]. In adolescents with ASD, DMN and right ECN hyperconnectivity, and SN and left ECN hypoconnectivity are associated with sensory deficits and impaired social communication [86,87,103]. In adults with ASD, hypoconnectivity between the DMN and VAN and between the SN and medial temporal lobe networks is strongly associated with impaired social functioning and reflects the severity of ASD [104,105,106,107]. Under normal circumstances, there is functional segregation between the primary somatosensory and auditory regions in children, which increases significantly with age. However, in children with ASD, there is hyperconnectivity between subcortical and cortical sensory regions that does not change with age. This reflects a delayed or arrested development of segregation in these areas, which is associated with socio-cognitive impairments [108]. In addition to the above-mentioned brain networks, researchers have found increased FC between the cerebellum and the posterior motor and somatosensory cortices, resulting in abnormal processing of sensory and motor functions [109].

In addition, Sha et al. [110] found reduced leftward lateralization in the fusiform, rostral middle frontal, and medial orbitofrontal cortices of children with ASD. There was also a decrease in rightward asymmetry of both degree centrality and global efficiency in the superior frontal cortex, indicative of node-level degree centrality asymmetries in children with ASD. These asymmetries are associated with executive function, working memory, and sensorimotor impairments. Floris et al. [111] found that compared with TD children, older children with ASD exhibited rightward lateralization in mean motor circuit connectivity, which may contribute to gross motor deficits and atypical gait. Previous studies have rarely explored ASD during infancy. Recent research suggests that key FC networks, such as the DMN and DAN, can be detected in infants at birth, which will aid in the early diagnosis of ASD [112]. Du et al. [113] combined network FC features in fMRI with gray-matter volume in sMRI for machine learning-based assessment of ASD, with an accuracy of 83.08%. Early diagnosis of ASD is based on fMRI deep learning, with an accuracy and sensitivity of 65.5% and 84%, respectively [2]. Combining fMRI with machine learning has further improved the diagnostic performance of ASD. Table 3 summarizes the fMRI-based literature on functional brain network damage in patients with ASD.

## 6. ASL-Based Alterations in ASD Perfusion

Arterial spin labeling is a non-invasive MRI technique that utilizes magnetically labeled arterial blood water as an endogenous tracer. It quantifies changes in cerebral blood flow (CBF) within functional brain regions, thereby reflecting the association between cerebral perfusion and core symptoms. Reduced CBF in children with ASD may lead to abnormal neuronal development in the brain, and the number of hypoperfused brain regions is positively correlated with age in children with ASD [104]. This abnormal neurodevelopment leads to cognitive, language, and motor developmental impairments in children with ASD [114]. However, CBF values in the frontal lobe show a non-linear correlation with age. At the ages of 2 and 5 years, CBF in the frontal lobe of children with ASD is normal. Around the ages of 3–4 years, there is a decline in CBF. This may be associated with slower growth and development of the frontal lobe in children with ASD starting at age 3, followed by a gradual normalization of frontal lobe development around age 5. However, after the age of 6, CBF in the frontal lobe gradually decreases [114]. The decreased CBF in the left frontal lobe, bilateral parietal lobes, bilateral temporal lobes, and insula in children with ASD is associated with impaired communication and socio-cognitive deficits and decreased self-care skills [115,116,117]. Reduced CBF in the bilateral fusiform and right inferior temporal gyri in adolescents with ASD is associated with deficits in social cognition and facial recognition [123]. Tang et al. [114] found that decreased CBF in the frontal, temporal, hippocampal, and caudate nucleus regions in individuals with ASD aged 2–18 years could be effective in the differential diagnosis of ASD, with the highest area under the curve (AUC) of 0.84. In addition, increased CBF in adolescents and adults with ASD reflects hyperperfusion, which may be related to the increase in metabolic demands caused by the remodeling of neural axons or glial cells after injury, with increased CBF in the medial orbitofrontal cortex, bilateral inferior frontal gyrus, and right precentral gyrus associated with social communication deficits in children [124,125]. In normal conditions, the FC within the brain of healthy individuals is tightly coupled with CBF and energy supply. This close coupling allows functional regions to engage in more neural activity, particularly in areas of the association cortex where the neurovascular coupling is higher, promoting active involvement in higher cognitive functions and efficient execution of effective information processing. At the same time, these regions exhibit a high long-range and a high CBF/FC ratio, reflecting the high metabolism of long-distance connections in the brain. In contrast, the primary visual cortex, which has abundant short-range connections, exhibits a lower CBF/FC ratio. It is speculated that the low metabolism associated with short-range connections helps conserve energy in the brain [126]. However, children with ASD may exhibit neurovascular decoupling in which decreased CBF in the anterior cingulate cortex is accompanied by increased FC associated with social cognitive deficits [124]. Children with ASD exhibit abnormalities in the inhibitory/excitatory balance of local neuronal clusters in the frontal cortex. The lack of inhibition may impair the establishment of long-range FC pathways and disrupt the balance between network metabolism and axonal connectivity, leading to neurovascular decoupling [124].

Therefore, altered neurovascular coupling may serve as an emerging biomarker for early diagnosis of ASD in children. A summary of the ASL-based ASD perfusion research literature is presented in Table 2.

## 7. Proton Magnetic Resonance Spectroscopy–Based Biochemical Metabolite Alterations in ASD

Proton magnetic resonance spectroscopy is a non-invasive neuroimaging technique used to quantify the concentrations of biochemical metabolites in specific regions of the brain. It reveals the pathological basis of ASD by identifying abnormalities in molecular behaviors. The primary metabolites assessed include *N*-acetylaspartate-containing compounds (NAA), creatine-containing compounds (Cr), choline-containing compounds (Cho), glutamate + glutamine (Glx), myo-inositol (mI), and gamma-aminobutyric acid (GABA). NAA, predominantly located within neurons and axons, serves as a biomarker for neuronal density, heterogeneity, and vitality [127,128]. Compared with TD children, children with ASD show a decrease in NAA concentrations. This decline primarily occurs in the left amygdala [129], bilateral orbitofrontal cortex [118], thalamus [130], anterior cingulate cortex [119,131], temporal cortex, cerebellum [127], and parietal lobe [132]. These reductions reflect impaired, diminished, or immature neuronal function in ASD, which is associated with social deficits and memory impairments [129,132]. Cr refers to the combined signal of creatine and phosphocreatine and plays a crucial role in maintaining energy homeostasis in the central nervous system. Children with ASD exhibit a reduction in the NAA/Cr ratio in the prefrontal white-matter region and anterior cingulate cortex, suggesting alterations in axonal function and cognitive impairments [119,120,128,133]. GABA, the predominant inhibitory neurotransmitter in the cerebral cortex, plays a crucial role in maintaining the balance of neural circuits [127]. Compared with TD children, children with ASD show decreased GABA levels in the frontal cortex, parietal lobe, and somatosensory motor area, reflecting excessive excitability in the cerebral cortex [121,127,134]. The decrease in GABA concentrations in the somatosensory motor area is associated with abnormal processing of tactile information. The reduced GABA concentration in the frontal cortex may be due to deficits in GABAergic interneurons, leading to glial activation, migration defects, and impairments in communication and cognition [127]. Furthermore, studies have found a positive correlation between GABA levels and age in the left parietal lobe of children with ASD [121]. In adults with ASD, there is an increase in GABA concentration in the left dorsolateral prefrontal cortex. The changes in GABA concentration in the prefrontal cortex, from decreased levels in childhood to increased levels in adulthood, reflect the importance of age-related alterations in brain neurometabolism in individuals with ASD [12]. Additionally, in children with ASD, there is a decreased GABA/Cr ratio in the left motor, anterior cingulate cortex, and auditory cortices, which is negatively correlated with the severity of ASD symptoms [122,135]. Glutamate (Glu), as an excitatory neurotransmitter, plays a crucial role in synaptic induction, cell migration, synaptic elimination, and other functions that are essential in neurodevelopmental processes. Glutamine (Gln) participates in the regulation of glutamate recycling and brain ammonia metabolism [127]. Glu, Gln, and GABA interact through the glutamate/GABA-glutamine cycle to maintain cortical excitatory/inhibitory balance, which is crucial for synaptic maturation, refinement of neuronal circuits, and regulation of cognition, emotion, and behavior [132,136]. Glx represents the overall levels of Glu and Gln and their functions in the brain. Compared with TD children, children with ASD have increased Glu concentration in the cingulate gyrus and prefrontal cortex, possibly due to decreased levels of glutamic acid decarboxylase, the enzyme that converts Glu to GABA in the brain [127,131]. Previous studies have suggested that an imbalance between excitation and inhibition in individuals with ASD forms the neurobiological basis of cognitive impairments [137,138]. ASD patients show alterations in GABA, Glx, Glu/Cr, and GABA+/Glu concentrations within the frontal and cingulate cortices at different developmental stages, reflecting an imbalance between neurotransmitter excitation and inhibition [12,139,140]. Cho, composed of phosphatidylcholine metabolites, is used to measure membrane turnover rate. mI, a key component of the second messenger system, serves as a specific marker for astrocytes [127]. The significant reduction of Cho and mI in the anterior cingulate cortex and frontal lobe of children with ASD indicates impaired neuronal integrity in these regions, which is associated with social impairments [119]. Hence, by quantifying abnormal changes in brain metabolites at different stages, ^1^H-MRS can assist in diagnosing ASD. In the future, the combination of ^1^H-MRS and multimodal MRI will help further delineate the diverse phenotypes of ASD.

## 8. Glymphatic System Changes in ASD

Evaluating the coupling relationship between neurovascular and cerebrospinal fluid (CSF) may reveal the complex pathophysiological mechanisms of the brain in ASD and may provide new insights into the early diagnosis of ASD. The glymphatic system is a unique network in the central nervous system of the brain, which allows the dynamic exchange of CSF and interstitial fluid through pathways such as the paravascular spaces (PVS). These play an important role in normal homeostasis and interstitial solute clearance. Water in the CSF can transport soluble Aβ and tau proteins and the energy metabolite lactate from the brain tissue through the induction of polarized astrocyte-specific aquaporin-4 into the interstitium [141]. Elevation in Aβ protein levels was observed in the neurons of postmortem brain tissue, blood, and peripheral CSF of individuals with ASD, which may have been associated with an impaired glymphatic system [142]. It was found that 44% of children with ASD had an enlarged PVS [143]. Recent studies have indicated that the function of the glymphatic system in the brains of individuals with ASD can be assessed using DTI along the perivascular space (DTI-ALPS). DTI-ALPS uses the diffusion tensor method to measure the diffusivity rate of water molecules and assess the movement of water molecules in the direction of the PVS. Studies have shown that reduced DTI-ALPS reflects impaired glymphatic function in children with ASD and is positively correlated with age [144]. In addition, children with ASD have increased extra-axial CSF (EA-CSF) at the age of 6–24 months and 2–4 years, which is associated with deficits in gross motor skills and non-verbal abilities [142,145,146]. Increased EA-CSF may be due to impaired early brain venous drainage or immature development of arachnoid granulations in young children with ASD, leading to reduced absorption [147]. Subsequently, the increased EA-CSF gradually normalizes after 4 years of age in children with ASD, which may be related to the gradual normalization of early increase in brain volume in later childhood [142,147]. Shen et al. [145] found that increased EA-CSF at 6 months of age could predict ASD at 24 months of age with 69% accuracy. Moreover, Diem et al. [148] suggested that arteriolar dilatation caused by neurovascular coupling could play a key role in the removal of cerebral waste products. Further studies are needed to determine whether neurovascular uncoupling involves the glymphatic system in children with ASD (Figure 3).

## 9. Conclusions

In summary, ASD is not only characterized by abnormal changes in brain morphology, structure–function connectivity, cerebral perfusion, and neuronal metabolism, but also by some degree of impairment in the function of the glymphatic system. Differences in age, subtype, brain damage, and remodeling in children with ASD, could lead to heterogeneity in research results.

## 10. Future Directions

Exploring the mechanism of brain damage in children with ASD and determining imaging biomarkers based on multimodal MRI remains the focus of future research, with an aim to provide an objective basis for the early identification of ASD and assist the clinic in formulation of individualized intervention plans. Multimodal MRI is expected to further assist in the early and accurate clinical diagnosis of ASD through deep learning combined with genomics and artificial intelligence.

## Figures and Tables

**Figure 1 diagnostics-13-03027-f001:**
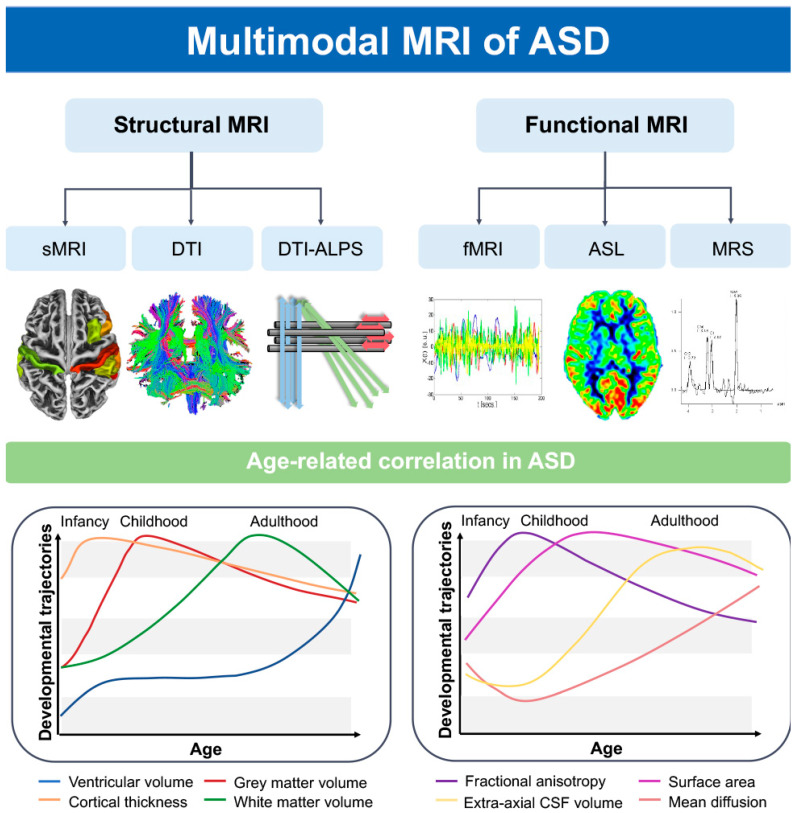
ASD patients’ brains exhibit distinct growth and developmental trajectories for various characteristics as age progresses [11,12]. Using structural magnetic resonance imaging (MRI), diffusion tensor imaging (DTI), functional MRI (fMRI), three-dimensional arterial spin labeling (3D-ASL), and proton magnetic resonance spectroscopy (^1^H-MRS), we can identify abnormal alterations in ASD patients’ brain morphology, structural-functional network, perfusion, neuronal metabolism, and the glymphatic system. FA, fractional anisotropy; MD, mean diffusion; EA-CSF, extra-axial cerebrospinal fluid; CT, cortical thickness.

**Figure 2 diagnostics-13-03027-f002:**
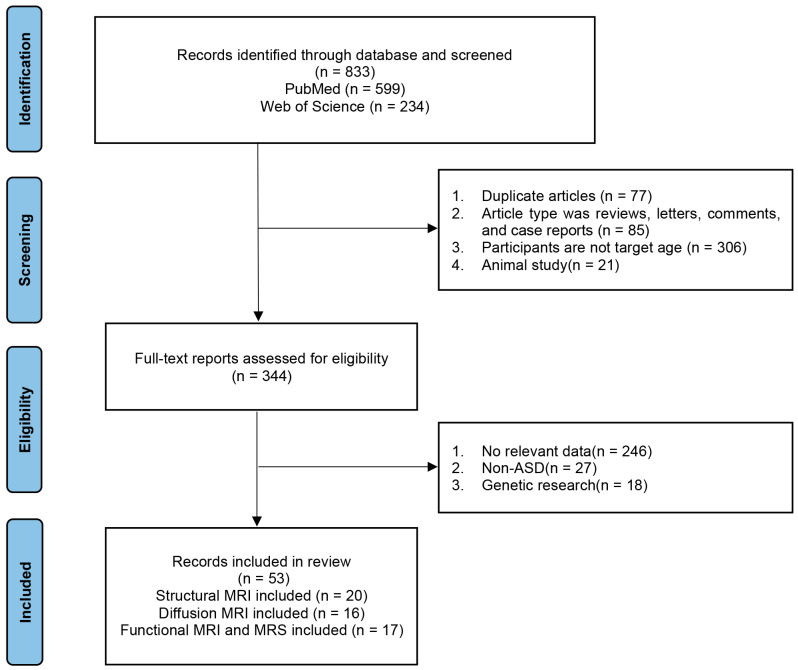
Flow chart of search results. ASD, autism spectrum disorder; MRI, magnetic resonance imaging; MRS, magnetic resonance spectroscopy.

**Figure 3 diagnostics-13-03027-f003:**
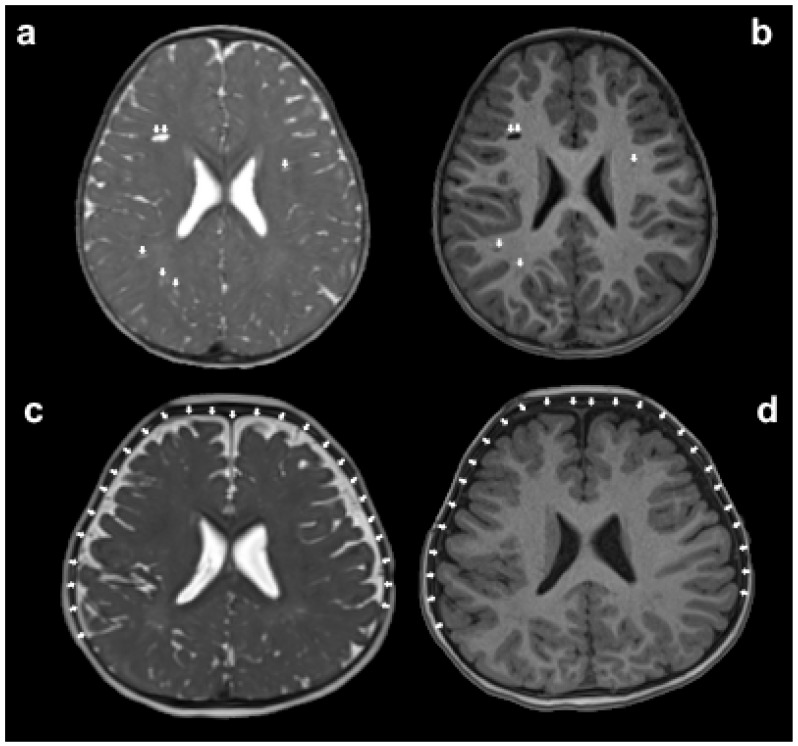
The assessment of glymphatic system disorders can be indirectly made by observing whether there is an enlargement in the volume of paravascular spaces (PVS) and extra-axial cerebrospinal fluid (EA-CSF) on magnetic resonance imaging (MRI). The white arrows in panels (**a**,**b**) indicate the enlarged PVS, while the white arrows in panels (**c**,**d**) point out the increased volume of EA-CSF.

**Table 1 diagnostics-13-03027-t001:** Structural magnetic resonance imaging of ASD.

Reference	Age Range (Mean)	Brain Regions	Main Findings in the ASD Group
Schumann et al. [6]	Longitudinal, 1.5–5 (2.5) years	Cerebral	↑ GMV and WMV in the cerebrum; notably in frontal, temporal, and cingulate cortices
Hazlett et al. [15]	(Longitudinal, prospective), 6–7, 12–13, 24–25 months	Cortical surface area	↑ Cortical surface area (6–12 month); ↑ TBV (12–24 month)
Hazlett et al. [17]	Longitudinal, 18–35 (32) month	Cerebral; CT	↑ Cerebral cortical (2 and 4–5 years); ↑ surface area in temporal, frontal, and parietal–occipital; no differences in cortical thickness
Ohta et al. [18]	Longitudinal, 2.5–3.5 (3) years	Cerebral cortical grey matter; surface area; cortical thickness	↑ Cortical surface; no difference in cortical thickness
Guo et al. [20]	2–7 (5) years	Cerebral; cerebellum	↑ GMV in fusiform face area and cerebellum/vermis
Bai et al. [21]	2–7 (5) years	Cerebral	↑ GMV in right medial superior frontal gyrus and left fusiform gyrus
Retico et al. [22]	2–7 (4.4) years	Cerebral	↑ GMV in bilateral superior frontal gyrus, precuneus, superior temporal gyrus; ↓ GMV in right inferior temporal gyrus
Li et al. [23]	6–24 months	Amygdala; hippocampus	↑ TBV in amygdala and hippocampus
Reinhardt et al. [24]	25–80 (38.2) month	Hippocampus	↑ TBV in hippocampus
Shou et al. [25]	2.9–5 (4.1) years	Amygdala; hippocampus	↑ TBV in left amygdala and left hippocampus
Mengotti et al. [26]	4–14 (7) years	Cerebral	↑ WMV in right inferior frontal gyrus, right fusiform gyrus, left precentral, supplementary motor area, and left hippocampus
Pote et al. [31]	4–6 (4.8) month	Cerebral	↑ TBV in cerebellar and subcortical
van Rooij et al. [37]	2–64 (15.4) years	Cerebral; CT	↓ Subcortical volumes of the pallidum, putamen, amygdala, and nucleus accumbens; ↑ CT in the frontal cortex; ↓ CT in the temporal cortex
Zielinski et al. [39]	3–36 (16.8) years	CT	↑ Thinning rate of CT in early childhood; ↓ thinning rate of CT in later childhood, adolescence, and early adulthood
Prigge et al. [41]	Longitudinal, 3.4–36 (16.4) years	CT	↑ CT in right frontal and temporal poles, and left superior frontal; ↓ CT in left inferior temporal and bank of the superior temporal sulcus, and right caudal anterior and posterior cingulate
Shiohama et al. [42]	10–35 (7.3) months	CT	↑ CT in right medial orbitofrontal cortex; ↓ CT in caudal anterior cingulate
Sussman et al. [43]	4–18 years	CT	↓ CT in left orbitofrontal cortex and left posterior cingulate gyrus
Postema et al. [44]	1.8–64 years	Cortical thickness asymmetries	↓ Cortical thickness asymmetries in medial frontal, orbitofrontal, inferior temporal, and cingulate regions
Yang et al. [49]	4–12 (8.4) years	Cerebral; cortical folding; CT; SA	↑ GI in right inferior parietal, inferior temporal, and the left isthmus cingulate gyri; ↑ CT and TBV in right middle temporal gyrus and the posterior superior temporal sulcus
Zoltowski et al. [53]	5–54 (15) years	Cortical folding; CT;	↑ lGI in right middle frontal gyrus, right inferior temporal gyrus, and right middle occipital; ↓ lGI in left posterior insula and right precuneus; ↑ CT in right anterior cingulate, right planum temporale/superior temporal gyrus

↑ Represents increase; ↓ represents decrease. ASD, autism spectrum disorder; TBV, total brain volume; GMV, gray-matter volume; WMV, white-matter volume; CT, cortical thickness; lGI, local gyrification index; SA, surface area.

**Table 2 diagnostics-13-03027-t002:** Diffusion magnetic resonance imaging of ASD.

Reference	Age Range (Mean)	Brain Regions	Main Findings in the ASD Group
Solso et al. [56]	12–48 (30.2) months	Frontal tracts	↑ FA and volume in forceps minor, inferior frontal superior frontal tract, uncinate, frontal projection of the superior corticostriatal tract
Ouyang et al. [57]	2–7 (4.1) years	Global main fiber tracts	↑ FA in most major white-matter tracts (before 4 years)
Andrews et al. [58]	Longitudinal, 2.5–7 years	Global main fiber tracts	↑ FA in middle and inferior cerebellar peduncles, superior longitudinal fasciculus, internal capsule, and splenium of the corpus callosum (young children); ↓ FA in sagittal stratum, cingulum, uncinate fasciculus, and internal capsule (63.6 months)
Andrews et al. [59]	25.9–49.6 (38.8) months	Global main fiber tracts	↑ FA in genu, body, and splenium of the corpus callosum, inferior frontal–occipital fasciculi, inferior and superior longitudinal fasciculi, middle and superior cerebellar peduncles, and corticospinal tract
Xiao et al. [11]	2–3 (2.5) years	Global main fiber tracts	↑ FA in corpus callosum, posterior cingulate cortex, and limbic lobes; ↓ MD in corpus callosum, posterior cingulate, limbic lobes, and insular cortex
Fu et al. [62]	2–9 years	Global main fiber tracts	↓ FA and ↑ MD in bilateral fornix, uncinate fasciculus
Hanaie et al. [63]	5–14 (9.8) years	Global main fiber tracts	↓ FA in bilateral superior cerebellar peduncle
Hrdlicka et al. [64]	5–13.2 (8.0) years	Global main fiber tracts	↓ FA in left arcuate fasciculus and inferior frontal occipital fasciculus
Lei et al. [65]	4–21 (9.3) years	Association fibers; projection fibers	↓ FA in association fibers (cingulum, inferior fronto–occipital fasciculus, inferior longitudinal fasiculus, superior longitudinal fasiculus, uncinate fasciculus), projection fibers (anterior thalamic radiation, corticospinal tract)
Weinstein et al. [66]	1.5–5.8 (2.9) years	Global main fiber tracts	↑ FA in genu of corpus callosum, and left superior longitudinal fasciculus
Fingher et al. [67]	13–51 (31) months	Global main fiber tracts	↑ FA in temporal corpus callosum segment
Walker et al. [70]	2–8 (4.7) years	Global main fiber tracts	↑ MD in posterior brain regions
Qin et al. [73]	2–6 (2.9) years	Topological network	↑ Nodal efficiency in left precuneus, thalamus, and bilateral superior parietal cortex
Li et al. [74]	3–6 (4.6) years	Topological network	↑ Nodal efficiency in left pallidum, right caudate nucleus; ↑ global efficiency and clustering coefficient; ↓ shortest path length
Qian et al. [75]	6–16 (9) years	Amygdala	↓ Nodal efficiency in right amygdala
Li et al. [78]	2–9 (3.9) years	Topological network	↓ Global efficiency; ↑ shortest path length

↑ Represents increase; ↓ represents decrease. ASD, autism spectrum disorder; FA, fractional anisotropy; MD, mean diffusion.

**Table 3 diagnostics-13-03027-t003:** Functional magnetic resonance imaging and proton magnetic resonance spectroscopy studies of ASD.

Reference	Age Range (Mean)	Brain Regions	Main Findings in the ASD Group
Haghighat et al. [90]	5–10 (7.3), 11–17 (13.7), 18–39 (25.9) years	Whole brain	↑ Connectivity in cingulate cortex, anterior insula, central opercular cortex, temporal pole, right anterior superior temporal gyrus, planum polare, middle frontal gyrus, right inferior frontal gyrus, cerebellum, and brainstem (children)
Xiao et al. [91]	1–4 (2.3) years	Temporal cortex	↑ Connectivity in temporal–cuneus, and temporal–precuneus
McKinnon et al. [92]	11–27 months	Whole brain	↑ Connectivity in DMN and control networks, DMN and DAN
Chen et al. [93]	17–45 (30) months	Whole brain	↑ Connectivity in visual and sensorimotor networks
Chen et al. [94]	3.5–7.9 (5) years	Whole brain	↑ Connectivity in sensory-motor and visual brain regions; ↓ connectivity in social cognition brain regions
Yerys et al. [100]	6–17 (12.4) years	Networks: VAN	↑ Connectivity in VAN and retrosplenial–temporal systems; ↓ connectivity in VAN and somatomotor-mouth systems
LLioska et al. [102]	5–58 (16) years	Networks: DMN; subcortical areas	↑ Connectivity in DMN and subcortex, ↓ connectivity in primary sensory and attention networks
Sha et al. [110]	2–64 (15.7) years	Whole brain	↓ Leftward asymmetry in rostral middle frontal, cuneus, medial orbitofrontal, and postcentral regions
Tang et al. [114]	2–18 years	Cerebral	↓ CBF in frontal lobe, hippocampus, temporal lobe, and caudate nucleus
Ye et al. [115]	3–8 (4) years	Cerebral	↓ CBF in left frontal lobe, the bilateral parietal lobes, and the bilateral temporal lobes
Mori et al. [116]	2–14 (7.3) years	Cerebral	↓ CBF in insula, superior parietal lobule, superior temporal gyrus, and inferior frontal gyrus
Tang et al. [117]	2–3 (2.7 years)	Cerebral	↓ CBF in frontal lobe, temporal lobe, hippocampus, caudate nucleus, substantia nigra, and red nucleus
Mori et al. [118]	3–6 (4) years	Amygdala and orbito-frontal cortex	↓ NAA in left amygdala and bilateral orbito-frontal cortex
Goji et al. [119]	2–12 (5) years	Anterior cingulate	↓ NAA, Cr, Cho, and mI in anterior cingulate cortex
Margari et al. [120]	1.7–14 (1.9) years	Frontal lobe	↓ NAA/Cr in frontal lobe white matter
DeMayo et al. [121]	4–12 (8.9) years	Parietal lobe	↓ GABA in left parietal lobe, ↑ GABA with age
Ito et al. [122]	2–15 (6.7) years	Anterior cingulate	↓ GABA/Cr in anterior cingulate cortex and left cerebellum, ↑ Glu/Cr in left cerebellum

↑ Represents increase; ↓ represents decrease. ASD, autism spectrum disorder; CBF, cerebral blood flow; DAN, dorsal attention network; VAN, ventral attention network; DMN, default mode network; NAA, N−acetylaspartate; Cr, creatine; Cho, choline-containing compounds; mI, myo-inositol; Glu, glutamate; GABA, gamma-aminobutyric acid.

## Data Availability

No new data were created during the writing of this review.

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
