# Peer review of "Application of Multimodal MRI in the Early Diagnosis of Autism Spectrum Disorders: A Review"

_diagnostics, 2023, doi:10.3390/diagnostics13193027_

Round 1
Reviewer 1 Report
Autism, a key form of brain pathology, remained for many years affected by various interpretations. For this the present review is largely interesting based on mechanistic results grown progressively during the last 20 years. The resulting framework includes comprehensive properties of autism spectrum disorders, important for the understanding of the diseases and the reinforcement of its diagnosis and treatments. As a whole, the presentation of the disease and its multimodal MRI interpretations are widely interesting and useful. However the Abstract is too short and superficial. It needs to become something such as a key tool of the enterprise, focused also on the heterogeneity of the ASDs. Analogous developments are needed for the Conclusions. A new form should be focused no only on the future, as it is now, but also on the critical properties and advantages of the general conditions.
The review is well conceived, including many important concepts and data. Upon consideration of specified aspects I will recommend the publication of this paper.
The English Language of the paper is appreciable. It could be further improved only at a high level, for example by the specific analysis from a scientist of mater language.
Author Response
Thank you for your comments and suggestions. We have added the response in the attachment.

Reviewer 2 Report
The work is well prepared. It would be good to make some additions.
1-How the literature review is done should be detailed.
2-A flow diagram should be added. Which databases were used?
3-What are the inclusion and exclusion criteria? Which keywords were searched?
Author Response

(The authors gave the same response as above.)
